# Inflammation and Digestive Cancer

**DOI:** 10.3390/ijms241713503

**Published:** 2023-08-31

**Authors:** Helge Waldum, Reidar Fossmark

**Affiliations:** Department of Clinical and Molecular Medicine, Faculty of Medicine and Health Sciences, Norwegian University of Science and Technology, 7030 Trondheim, Norway; reidar.fossmark@ntnu.no

**Keywords:** carcinogenesis, inflammation, chronic gastritis, gastric cancer, chronic hepatitis, hepatocellular carcinoma, inflammatory bowel disease, colon cancer, mechanism, hormonal carcinogenesis

## Abstract

Chronic inflammation is linked to carcinogenesis, particularly in the digestive organs, i.e., the stomach, colon, and liver. The mechanism of this effect has, however, only partly been focused on. In this review, we focus on different forms of chronic hepatitis, chronic inflammatory bowel disease, and chronic gastritis, conditions predisposing individuals to the development of malignancy. Chronic inflammation may cause malignancy because (1) the cause of the chronic inflammation is itself genotoxic, (2) substances released from the inflammatory cells may be genotoxic, (3) the cell death induced by the inflammation induces a compensatory increase in proliferation with an inherent risk of mutation, (4) changes in cell composition due to inflammation may modify function, resulting in hormonal disturbances affecting cellular proliferation. The present review focuses on chronic gastritis (*Helicobacter pylori* or autoimmune type) since all four mechanisms may be relevant to this condition. Genotoxicity due to the hepatitis B virus is an important factor in hepatocellular cancer and viral infection can similarly be central in the etiology and malignancy of inflammatory bowel diseases. *Helicobacter pylori* (*H. pylori*) is the dominating cause of chronic gastritis and has not been shown to be genotoxic, so its carcinogenic effect is most probably due to the induction of atrophic oxyntic gastritis leading to hypergastrinemia.

## 1. Introduction

Neoplasia is a type of abnormal cell growth due to an imbalance between cell division and cell death caused by changes in growth control. In other words, there is always a dysregulation of cellular turnover in a neoplasia manifesting itself as a tumor. Moreover, it follows that only cells having the ability to divide/proliferate can give rise to neoplasia. Malignant neoplasms can grow beyond their natural bounds (invasion) and settle in discontinuity with the tumor (metastasis), whereas benign tumors lack these properties. There have been different theories of the cells of origin of neoplasia regarding whether they originate from stem cells that stop their development at a certain stage or they develop from mature cells that dedifferentiate [1,2]. The stem cell origin of malignant tumors has been the prevailing theory during the last few decades [3]. However, it is evident that also mature cells may develop into malignant tumors, as seen in the gastric oxyntic mucosa, where continuous gastrin hyperstimulation causes the enterochromaffin-like (ECL) cell to proliferate and, through stages of ECL cell hyperplasia via a neuroendocrine tumor (NET), become a highly malignant carcinoma, finally killing the patient [4]. The correct identification of the cell of origin and its implications have been discussed previously in connection with the famous Japanese pathologist Soga [5]. 

Neoplasms are the result of cellular genetic changes that affect growth regulation and other cellular properties. The genetic changes may be congenital but are most often due to acquired mutations, which may occur by chance as the result of cell division or induced by chemicals or irradiation (mutagens). Stimulated cell division to replace damaged/lost cells and/or stimulation by hormones or chemicals (mitogens) will necessarily also induce an increased risk of mutation.

Chronic inflammation is presently accepted as an important mechanism for carcinogenesis causing hepatocellular carcinoma in patients with hepatitis, cancer of the colon in inflammatory bowel disease (particularly ulcerative colitis) and gastric cancers secondary to gastritis as the most typical examples [6]. In this review, we will discuss the mechanisms involved in carcinogenesis by chronic inflammation in general. Inflammation is a very complex process involving a variety of different cells and a multitude of molecules. The innovative aspect of this review will be to assess whether the cause of the inflammation is directly carcinogenic by itself, whether the carcinogenesis is a consequence of the cell destruction resulting in accelerated proliferation and whether the tissue changes (scarring) resulting in changes particularly in the concentrations of mediators regulating proliferation, are involved. The knowledge gained from life-long research in gastric physiology and pathology and presented in many publications in the last decade is here extended to other digestive organs with respect to carcinogenesis.

## 2. The Mechanisms of the Carcinogenic Effects of Chronic Inflammation

Inflammation occurs as a response to exogenous agents like microorganisms or toxins or to direct trauma. Acute inflammation is initiated by tissue-resident immune cells that, upon activation, attracts neutrophilic granulocytes and macrophages. Lymphocytes (T and B cells) are pivotal in the further inflammatory response and start the defense to remove the exogenous agent and prepare for healing [7]. Most threats are contained and eliminated by the acute inflammatory response by phagocytosis and the release of substances that destroy infectious agents, but sometimes chronic inflammation develops when the exogenous agent persists, leading to chronic disease. In many chronic inflammatory diseases, the exogenous agent is not known, which may have led to the concept of autoimmunity. However, there are reasons to doubt that defaults in our immune apparatus are the main causes of many of our most important diseases. It appears more probable that we have not yet identified the foreign agent stimulating the immune cells. Although chronic inflammation is associated with cancer development, there exist few convincing studies indicating a cancer-preventive effect of non-steroid anti-inflammatory agents (NSAIDs) [8]. The anticancer effect of NSAIDs is believed to be due to their inhibition of the cyclo-oxygenase, particularly cyclo-oxygenase type 2 [9]. A recent study did not confirm the previous belief that non-aspirin NSAIDs have a protective effect on gastric cancer after *H. pylori* eradication [10]. Furthermore, the cyclo-oxygenase inhibitor 2 celecoxib did not affect the prognosis in cancer when given together with standard chemotherapy [11]. Despite the many years that NSAIDs have been claimed to have a positive effect on cancer, their role in cancer treatment has not been substantiated clinically since there are no randomized studies that support results from observational or epidemiological studies. Glucocorticoids also have anti-inflammatory effects, but long-term use is associated with a higher risk of several cancers, including liver cancer [12]. The lack of a convincing cancer-preventive effect of the treatment of chronic inflammatory diseases with corticosteroids or NSAIDs does not support the view that inflammation is directly carcinogenic, but rather that the secondary changes that it induces predispose patients to cancer. However, so-called autoinflammatory diseases related to the innate immunity not involving T and B immune cells are seldom but do exist, where there seems to be no exogenous cause of the inflammation. Autoinflammation is often hereditary [13] and is not known to play any role in the chronic inflammation observed in other diseases. 

### 2.1. Chronic Hepatitis 

Chronic viral hepatitis caused by either hepatitis B virus (HBV) or hepatitis C virus (HCV) both predispose patients to liver cirrhosis as well as hepatocellular carcinoma (HCC). However, HBV, being a DNA virus, integrates into the host genome and thereby contributes to hepatocellular carcinogenesis, besides the inflammation itself [14], which probably is the sole mechanism for the carcinogenesis caused by HCV [14]. In children infected with HBV perinatally, fewer have cirrhosis in combination with HCC than those infected later in life [15]. Human papilloma virus (HPV) causing cervicitis, as well as cervix cancer, is an example of a virus like HBV that causes cancer via a genotoxic effect, as well as chronic inflammation [16]. For HCC caused by HBV, the inflammation may not be the main cause of the neoplasia, but the inflammation and the neoplasia are the consequences of the infectious agents themselves. Antiviral therapy without maintenance of a virologic response implies an increased risk of HCC compared with those with a maintained response [17], and residual viraemia during antiviral therapy is connected to HCC [18]. Thus, the virus and not the inflammation is central in HBV-driven hepatic carcinogenesis. The increased risk of HCC in patients with steatohepatitis [19], in contrast to those with hepato-steatosis [20], on the other hand, demonstrates that inflammation itself may be the sole carcinogenic factor. However, the risk of HCC in patients with cirrhosis due to viral hepatitis is higher than in patients with cirrhosis due to other liver diseases [21]. Loss of hepatocytes leading to increased proliferation is the most probable mechanism for the neoplasia in steatohepatitis and possibly also HCV hepatitis. HCV infection is now treated with direct activating antivirals, leading to the eradication of the virus (sustained viral response (SVR)), resulting in a reduced risk of HCC in non-cirrhotic and cirrhotic livers [22]. However, depending on the degree of liver abnormality (fibrosis, cirrhosis), the risk of HCC persists in patients with HCV even after virus eradication [23,24]. This may be explained by the fact that carcinogenesis is a multistage process that progresses with time. At the time of HCV eradication, liver cells may be at different stages of transformation and a proportion may spontaneously develop further. Liver fibrosis as well as the extracellular matrix (ECM) may vary between stages of liver disease, possibly via the activation of hepatic stellate cells [25], but their role in carcinogenesis is not completely clarified. The effect of acetylsalicylic acid on the risk of HCC has been studied in a series of observational studies and an overall risk reduction in the general population has been found in a recent meta-analysis [26]; this included studies of sub-populations with HBV and HCV. However, in a meta-analysis, there may be publication bias, and randomized studies are indicated to clarify the true effects.

### 2.2. Inflammatory Bowel Disease

The most prevalent form of colonic cancer develops via adenomas polyps, whereas the cancers related to inflammatory bowel disease (IBD) start as flat dysplasia. Moreover, genetic changes also differ between these two types of colonic cancer [27]. Ulcerative colitis (UC) predisposes patients to cancer of the colon after long-term disease. This relationship has been recognized for decades [28] and is one of the strongest arguments for an important role of inflammation in carcinogenesis. The mechanism of this carcinogenesis is not known, which is not surprising since the cause of ulcerative colitis is also unidentified. However, the cancer risk is increased compared to the background population in UC patients in whom most of the colon is affected (total colitis), whereas the risk increase is neglectable in patients with inflammation restricted to the rectum [29]. Patients with primary sclerosing cholangitis (PSC) accompanying UC are particularly at risk of colonic cancer localized to the proximal colon, which may show only subclinical inflammation [30]. The fact that UC predisposes patients to colonic cancer and PSC to cholangiocarcinoma indicates that the cause of inflammation and not local factors is carcinogenic. Although colon cancer may develop in patients after decades of inactive UC, there is agreement that chronic active inflammation increases the risk [31,32,33]. The cause of UC could, for instance, be a virus [34,35], which could itself contribute to the carcinogenesis. Interestingly, bacteriophages have also been discussed in the etiology of IBD [36]. Moreover, patients with Crohn’s disease (CD) have an increased risk of colorectal cancer when the inflammation affects this organ, although at a lower risk than in those with UC [37]. Nonetheless, since CD may have a segmental distribution with frequent involvement of the mesentery, the etiology of the inflammation could be connected to the nerves, probably infection of the ganglions [34]. A role of the mesentery with its nerves and blood vessels [38], as well as the enteric glial cells [39], has been discussed in the pathogenesis of CD. The onset of IBD in the elderly does not imply any increased risk of colonic cancer [40], possibly due to the latency of more than 8–10 years between the onset of colitis and the diagnosis of cancer [41]. The relatively recent strategy of inducing “deep” or histological remission is interesting, but its effect on carcinogenesis cannot be evaluated over several years. Finally, concerning the etiology, the association of pyoderma gangrenosum [42] and alopecia areata [43] with IBD may hopefully in the future be explained.

Microscopic colitis is a more recently defined disease characterized by chronic inflammation in the colon. There has been interest in the risk of colon cancer. Despite long-term inflammation, the risk of colon cancer seems to be decreased or similar to the background population in numerous epidemiological studies [44]. Thus, the increased risk of colonic cancer in patients with chronic long-term inflammation depends on the cause of inflammation, which unfortunately is not yet known.

### 2.3. Chronic Gastritis and the Carcinogenic Effect of Helicobacter pylori

For clarity, the distribution of the cell types central in gastric carcinogenesis, as well as the regulation of gastric acid secretion, is depicted in Figure 1.

Gastric cancer may be caused by inborn errors [46,47] as well as by Epstein–Barr virus [48] belonging to the herpes group, known to be generally carcinogenic due to the incorporation of its DNA into the human genome [49]. Gastric cancer secondary to Epstein–Barr virus has been reported to occur more often in males and have a proximal location [50]. However, the main cause of gastric cancer is the bacterium *H. pylori*, which, as the first bacterium, was recognized as a class 1 human carcinogen by the WHO in 1994. In the next paragraph, we will discuss the mechanism of the carcinogenic effect of *H. pylori* infection in detail. *H. pylori* infection induces acute inflammation, often with a short symptomatic period, in the antrum, spreading to the oxyntic area by establishing itself in the mucous layer. After a short period, the symptoms as well as the acute inflammation subside. Nevertheless, a chronic infection persists in the antrum, which, over years, spreads to the oxyntic mucosa, where it induces atrophy, which finally can lead to the complete destruction of the glandular mucosa [51].

Gastric cancer was previously one of the most prevalent types of cancer. Although its frequency has declined substantially during the last few decades due to the decline in Laurén [52] intestinal-type cancers, the stomach is still an important site of cancer localization due to high mortality, which has shown only a slight reduction. Moreover, there are great geographical differences in the occurrence of gastric cancer [53]. Gastric cancer is unique in that a bacterial infection, *H. pylori*, causes gastritis [54], which is the dominating factor in subsequent carcinogenesis [55]. Since *H. pylori* may be eradicated by treatment with a combination of antibiotics and drugs, markedly reducing gastric acidity, this opens opportunities for the prevention of gastric cancer. However, the relationship between *H. pylori* gastritis and gastric cancer is complex, since *H. pylori* does not predispose patients to cancer before having induced oxyntic atrophy [56]. When the gastritis only involves the antral mucosa, it predisposes patients to duodenal ulcer but protects against gastric cancer development [57]. Moreover, autoimmune gastritis (AG), which affects only the oxyntic mucosa, leading to atrophy, also predisposes patients to gastric cancer [58,59], further strengthening the relationship between oxyntic atrophy and gastric cancer. There is no doubt that inflammation causes the oxyntic atrophy both in *H. pylori* and autoimmune gastritis. Thus, oxyntic gastritis leading to oxyntic atrophy is common to the two most important causes of gastric cancer, which suggests that *H. pylori* does not have a specific carcinogenic effect besides causing inflammation leading to gastric atrophy. A recent attempt to indicate that *H. pylori* infection could have occurred at an early phase of autoimmune gastritis [60] was not persuasive [61]. A paper by Usui et al. [62] with a commentary [63] in the New England Journal of Medicine described that germline pathogenic variants of known cancer-predisposing genes contributed to gastric cancer by having an additive effect on *H. pylori* carcinogenesis. Carcinogenic factors are expected to interact, but the authors found that this was particularly true for homologous recombination genes and speculated that the carcinogenic effect of *H. pylori* gastritis was due to a genotoxic effect of CagA, leading to errors in DNA repair [62]. The most important finding of their study was the demonstration of a continuous functional risk in carcinogenesis based on the genetic structure, explaining the well-known high occurrence of cancer in some families hitherto without an established genetic cause. The additive carcinogenic effect of the genetic variants and the *H. pylori* infection is, however, not proof of a direct genotoxic effect of *H. pylori*. In the Usui et al. study, *H. pylori* infection was assessed by *H. pylori* antibodies and reduced pepsinogen I (PG I) as a marker of atrophic gastritis presumed to be due to *H. pylori* infection [62]. Unfortunately, PG I reduction does not discriminate between oxyntic atrophy caused by *H. pylori* and autoimmune gastritis. Moreover, serum PG I is not a sensitive marker of oxyntic atrophy since incomplete atrophy with some remaining chief cells will lead to normal values [64], which may explain the surprisingly low frequency of pathological serum PG I in patients with *H. pylori* infection and gastric cancer [62]. Preferably, oxyntic atrophy should be assessed by gastrin elevation, which also would allow the evaluation of the role of gastrin versus *H. pylori* in combination with the genetic variations. A direct carcinogenic effect of *H. pylori* does not explain the essential role of oxyntic atrophy in *H. pylori*-induced gastritis [56] or that antral gastritis causing duodenal ulcer protects against gastric cancer [57]. Moreover, as written in a review in 2014, “Despite a close causal link between H pylori infection and the development of gastric malignancies, the precise mechanisms involved in this process are still obscure” [65]. Interestingly, patients having lost *H. pylori* at a stage of oxyntic atrophy still have an increased risk of developing gastric cancer decades after having lost *H. pylori* [66]. *H. pylori* gastritis seems not to predispose patients to gastric cancer of the cardia [67,68]. Besides gastric cancer, *H. pylori* gastritis also predisposes patients to gastric lymphoma of the so-called mucosa-associated lymphoid tissue (MALT) [69]. 

#### 2.3.1. *H. pylori* Virulence and Direct Carcinogenic Effects

With the acceptance of *H. pylori* as a class 1 carcinogen, the WHO indirectly recognized that *H. pylori* was itself carcinogenic, causing not only gastric carcinoma but also MALT lymphoma. Inflammation has been claimed to induce DNA damage [70] and the immune microenvironment to be important in gastric carcinogenesis [71], although this has not been shown experimentally. The fact that antral affection apparently protects against gastric cancer and that there is no increase in cardia cancer does not suggest a direct carcinogenic effect. The extensive and long-term negative search for a direct carcinogenic factor of *H. pylori* supports this view. Nevertheless, Cag pathogenicity island positivity (CagA+) increases the risk of gastric cancer [72,73], although *H. pylori* CagA negative (CagA−) also predisposes patients to gastric cancer [74]. The mechanism of the increased risk of gastric cancer when infected with CagA+ *H. pylori* was recently reviewed, with the conclusion that there is much to be clarified [75]. Of interest is the difference in CagA motives between people from Western and Eastern countries, the latter with high incidence of gastric cancer [76,77]. Interestingly, CagA positivity increases the severity of the inflammation [78] and the risk of atrophic gastritis [79], further supporting the connection between *H. pylori*, atrophic gastritis and gastric cancer. On the other hand, *H. pylori* gastritis predisposes patients to gastric MALT lymphoma independently of the occurrence of atrophic gastritis [80], indicating different pathogeneses for these two gastric malignancies secondary to *H. pylori* gastritis. Furthermore, this fits well with the fact that there is no report of increased gastric MALT lymphoma in patients with autoimmune gastritis. Moreover, CagA status does not play any separate role in the pathogenesis of gastric MALT lymphoma [81]. However, not only autoimmune gastritis [82] but also *H. pylori* gastritis [83,84] predisposes patients to gastric NETs. The higher incidence of NETs in autoimmune gastritis compared with *H. pylori* gastritis may be explained by the more rapidly developing complete oxyntic atrophy and antral sparing, resulting in more pronounced hypergastrinemia [85]. It may accordingly be concluded that it is not definitively shown that *H. pylori* is a direct carcinogen.

#### 2.3.2. Inflammatory Host Factors

It is evident that *H. pylori* gastritis is a major cause of gastric cancer. Since no direct carcinogenic factor has been identified in *H. pylori*, El-Omar and co-workers studied whether there were genetic differences in the inflammatory response between those with and without cancer. Interleukin-1 gene cluster polymorphisms presumed to increase interleukin-1 beta were associated with both reduced gastric acidity and gastric cancer [86]. Thus, genetic host factors did not distinguish between the consequences of gastritis, hypoacidity and gastric cancer. The relationship between genetic differences in inflammation and gastric cancers was later substantiated [87,88,89]. A genetic polymorphism of the interleukin-8 gene has also been reported to affect the risk of the severity of oxyntic gastritis as well as gastric cancer [90]. Of great relevance to the role of cytokines in the development of cancer was a recent description of a reduction in lung cancer after the inhibition of interleukin-1beta [91]. Neither by differences in *H. pylori* nor host inflammatory factors is it thus possible to sever the connection between inflammation and gastric cancer.

#### 2.3.3. Accelerated Cell Death, Proliferation and Mutation Risk Caused by Inflammation

Increased cell division was suggested as an important cause of human cancer many decades ago [92]. Furthermore, increased gastric mucosal proliferation without an accompanying increase in apoptosis has been found to be correlated with the severity of gastritis in CagA and VacA *H. pylori*-infected subjects [93]. Tomasetti and Vogelstein have hypothesized that the variation in cancer risk between different tissues could be explained by the number of cell divisions of the stem cells [94]. Similarly, an important mechanism of the carcinogenesis of *H. pylori* could be the stimulation of proliferation to replace cells lost due to cell destruction secondary to inflammation. Such a mechanism may be the same for carcinomas as well as MALT lymphomas. However, it does not explain the fundamental role of oxyntic atrophy in gastric carcinogenesis [56] or the occurrence of gastric carcinomas in patients with oxyntic atrophy decades after the loss of *H. pylori* and gastritis [66], which is best explained by the consequences of gastric hypoacidity.

#### 2.3.4. Lack of Acid Secretion

The main function of the oxyntic mucosa is to produce acid and the enzyme pepsin, both participating in the killing of swallowed microorganisms and thus preventing many infections [95]. Pepsin activity is lost at a pH above 4.0 [96], the same value at which bactericide activity is abruptly reduced [97]. The most important regulator of gastric acidity is gastrin [98], produced by the G-cell located in the antral mucosa. The G-cell senses directly or indirectly, via the somatostatin producing D-cell [99], the H+ concentration of the gastric content and adjusts its gastrin release accordingly in order to maintain adequate acidity. Hypoacidity secondary to oxyntic atrophy will therefore take away a very important biological function, the killing of swallowed microorganisms, which is, in this situation, in vain, counteracted by maximal gastrin release. If *H. pylori*-induced oxyntic atrophy should predispose patients to gastric cancer via changes in microbes, one would expect similarly increased occurrence throughout the stomach. However, *H. pylori* does not increase the frequency of cancer localized to the cardia [100]. Curiously, although there are different mucosae in the corpus/fundus and antrum, it has not been possible to discriminate between gastric cancers localized to these mucosae. This may partly be due to fact that the border between the oxyntic and antral mucosae is not sharp, with the oxyntic glands found in the antrum [45]. Nevertheless, we found that patients with hypergastrinemia had an increased risk of gastric carcinoma only localized to the oxyntic mucosa [101]. However, it is possible that the viral contamination of a stomach without acid also could have a carcinogenic effect. Oxyntic atrophy with gastric hypoacidity leads to hypergastrinemia, known since the early days of gastrin immunoassays [102].

#### 2.3.5. The Role of Gastrin in Gastric Carcinogenesis

Hypoacidity was recognized early on to be connected to gastric cancer [103,104]. Similarly, gastritis was known to predispose patients to gastric cancer [105,106]. Thus, the description of *H. pylori* as the most important cause of chronic gastritis [54] and gastric cancer [55] connected these factors together. Approximately 70 years ago, neuroendocrine differentiation in gastric cancers was described [107], and approximately 50 years ago, it was reported that the ECL cell gives rise to the NETs occurring in patients with atrophic gastritis with pernicious anemia [108]. In the middle of the 1980s came reports describing ECL-derived oxyntic tumors in rodents after long-term dosing with efficient inhibitors of gastric acid secretion belonging to the proton pump inhibitor (PPI) group [109] or being an insurmountable histamine-2 receptor blocker [110] giving rise to the gastrin hypothesis [111]. Interestingly, ECL cell differentiation occurs rather commonly in gastric carcinomas also in humans and particularly in the diffuse type [112,113]. By applying immunohistochemistry with increased sensitivity using tyramide signal amplification, we detected that more gastric cancers overall displayed neuroendocrine differentiation [114] and that virtually all cancers related to hypergastrinemia did so [115]. Gastric cancer classified as signet ring cell cancer (a subgroup of cancers of diffuse type) was shown to express neuroendocrine markers including ECL cell-specific markers [116] but not mucin [117]. Interestingly, signet ring cell carcinomas have been described in patients with multiple endocrine neoplasia type I (MEN1) with gastrinoma, indicating an important role of gastrin in the pathogenesis [118]. Furthermore, the gastrin receptor may be detected by immunohistochemistry or in-situ hybridization in an important proportion of gastric cancer cells [119]. It may accordingly be concluded that the target cell of gastrin, the ECL cell, plays an important role in gastric carcinogenesis via gastrin [120].

Gastrin is a peptide hormone and can directly only affect cells expressing its receptor, but indirectly it can also influence other cells via stimulation of the release of signal substances such as the regenerating (Reg) protein from ECL cells [121]. The Laurén classification of gastric carcinomas into intestinal and diffuse types seems to reflect an important biological difference since the two types do not differentiate into one another [52]. Moreover, the decline in the occurrence of gastric cancer seen during the last few decades is mainly due to a fall in the incidence of the intestinal type [122]. To our knowledge, our explanation of the role of gastrin in stimulating the ECL cell directly and thereby predisposing patients to gastric carcinomas of the diffuse type, and the stem cell via the stimulation of the release of mediators from the ECL cell leading to gastric carcinomas of the intestinal type, is presently the only plausible theory of gastric carcinogenesis secondary to oxyntic atrophy (Figure 2) [123,124]. Moreover, every condition with long-term hypergastrinemia in humans causes gastric neoplasia, mainly carcinomas of ECL cell origin [47,115,125,126]. Gastric carcinoma in patients with gastrinoma has, on the other hand, rarely been described, whereas ECL cell NETs occur often in patients with gastrinoma, whether being secondary to MEN1 [127] or of the spontaneous type [128,129]. A case report describing that the removal of all gastrinomas in a patient with MEN1 caused the gastric NETs to disappear showed that gastrin is the dominating cause of these tumors also in this condition [130]. The reason that only a few gastric carcinomas in patients with gastrinoma have been described is probably the short lifespan with hypergastrinemia in these patients, taking into consideration the long latency required for cancer to develop [47]. The important role of gastrin in human carcinogenesis is also strongly supported by the fact that long-term hypergastrinemia predisposes patients to gastric neoplasia in all species examined [131,132]. Thus, gastric hypoacidity stimulates gastrin release, which stimulates the release of signal substances including histamine and Reg protein [121] from the ECL cell, which, at the same time, is stimulated to divide, thereby increasing the risk of neoplasia (NETs and carcinomas of diffuse type), whereas the signal substances from the ECL cell stimulate the stem cell towards proliferation and neoplasia [123]. The ECL cell is situated at the bottom of the oxyntic glands, close to the basal membrane, and avoids destruction during gastritis. The expression of ECL cell markers including the gastrin receptor in gastric cancers strongly supports the dominating role of gastrin in gastric carcinogenesis.

The role of the different mechanisms of inflammation in carcinogenesis of the major digestive organs is shown in Table 1. 

## 3. Conclusions—Chronic Inflammation and Cancer

Chronic inflammation is present in several organs in the gastrointestinal tract in conditions associated with an increased risk of cancer. Inflammation seems to be a phenomenon paralleling or being secondary to more important processes in the liver, colon, and stomach. Like other bacteria, *H. pylori* is not a direct carcinogen. The carcinogenesis due to *H. pylori* gastritis is a consequence of the damage caused by the destruction of the oxyntic mucosa secondary to the inflammation that it induces, and therefore inflammation and carcinogenesis will be closely connected between different strains. The hypoacidity due to oxyntic atrophy induces hypergastrinemia, which stimulates the ECL cell to release signal substances as well as to proliferate. The stimulation of the ECL cell’s proliferation predisposes patients to ECL cell-derived neoplasia, including ECL cell NETs and gastric carcinomas of the diffuse type, whereas the stimulation of the release of ECL cell signal substances including Reg protein predisposes patients to carcinomas of the intestinal type via their stimulation of stem cell proliferation.

## Figures and Tables

**Figure 1 ijms-24-13503-f001:**
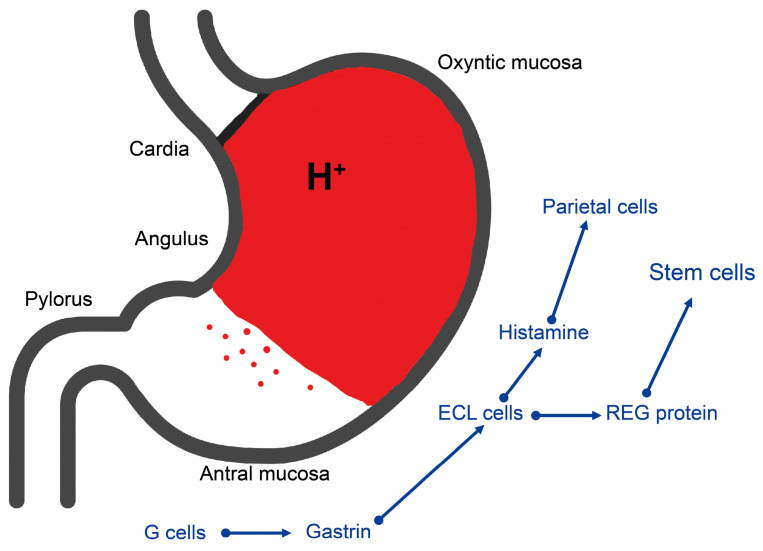
The different mucosae of the stomach with the central regulatory and secretory cells also involved in tumorigenesis. The red dots indicate the oxyntic gland in the antral mucosa according to Choy et al. [45].

**Figure 2 ijms-24-13503-f002:**
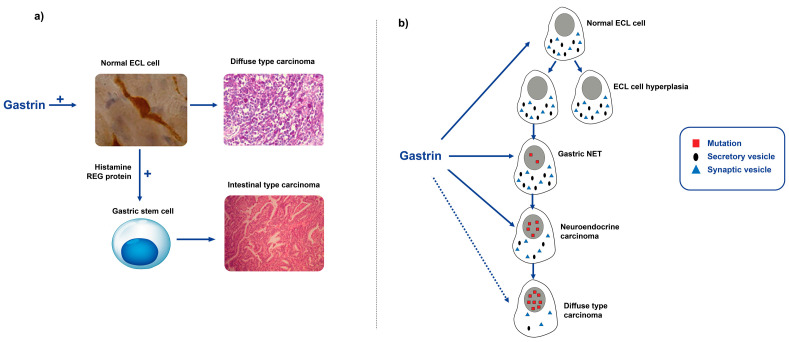
The central role of gastrin and the ECL cell in gastric carcinogenesis due to oxyntic atrophic gastritis (with permissions (**a**) from [125] and (**b**) from [133]).

**Table 1 ijms-24-13503-t001:** The mechanisms for tumorigenesis/carcinogenesis related to inflammation in the major digestive organs.

Organ	Pathogenesis	Disease/Agent	Mechanism of Carcinogenesis		
			Causative Agent Is Carcinogenic	Increased Proliferation	Secondary Hormonal Changes
Liver		HBV	++	+	
		HCV	?	++	
		Steatohepatitis		++	
Colon		Ulcerative colitis	?	+	
		Crohn’s disease	?	+	
Stomach	Oxyntic atrophy	Helicobacter pylori gastritis		+	++
		Autoimmune gastritis		+	++
		Epstein-Barr virus	+	+	

+ and ++: degree of involvement in carcinogenesis; ?: uncertain or possible.

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
