# Peer review of "Inflammation and Digestive Cancer"

_ijms, 2023, doi:10.3390/ijms241713503_

Round 1

Reviewer 1 Report

A review about “Inflammation and Cancer,Several points should be noted as below.

1) The title Inflammation and Cancerneeds to be changed, because Cancerhere only includes gastrointestinal-tumor.

2) How about the diversity and role of inflammation-related immune cells in gastric cancer/HCC/colon cancer?

3) As to “2.1. Chronic hepatitis”, how about the reciprocal role of ECM and fibrosis in liver cancer.

4) How about the role of EBV in gastric carcinogenesis?

5) The table 1 about “ The mechanisms for the

tumorigenesis/carcinogenesis related to inflammation in the major digestive organs.”was too simple. As least, the molecular details about inflammation in gastric carcinogenesis need to be constructed in the form of Figure.

6) A recent paper shows that cancer is an ecological disease that is  a multidimensional spatiotemporal ecological/-evolutionary process as a whole.  (https://www.thno.org/v13p1607.htm). It should be helpful for understating the cancer and TME.

Author Response

Thank you for a kind evaluation.

We have made the following changes.: 1: The title changed according to suggestion. :2 and 3. There are many papers focusing on the role of different immune cells in carcinogenesis, but they have not resulted in any firm conclusion. Moreover, the present review gives arguments against any direct carcinogenic effect by inflammation .In stead , the carcinogenesis seems to be secondary to the cell destruction caused by the inflammation. Therefore, we do not find it natural to discuss the different cells with respect to carcinogenesis. However, we added a sentence and a new reference (25) related to ECM and fibrosis. Lines123-126.

4. Thank you for a very valuable suggestion, Epstein-Barr virus. EBV is is shortly discussed on Lines 184-187, added three new reference, 48-49-50 and EBV is also included in the table.

5. A new figure (1) is included which should make it easier to grasp the ideas behind this review. In the figure also some central molecules are also included.

6. The paper described by the referee is certainly interesting, but we felt it to be beyond the scope of this paper, and we have therefore no commented on it.

Reviewer 2 Report

Topic of manuscript is interesting and suitable for IJMS. Strong point of manuscript are discussed clinical trials. Nevertheless, for the higher impact and illustrativeness of manuscript, I can recommend incorporate table of clinical trials into manuscript.

In my opinion, the manuscript does a good job of introducing the reader to the issues without overwhelming him with details. On the other hand, this approach can strongly decrease the range of issues described. The text is more suitable for a general publicum than specialist in specialists in the field.  Because adding detail directly to the text could reduce its fluency I suggest adding tables to the manuscript. I consider the most important table of clinical studies which should contain enough detail to get a more detailed overview of the studies in question. Table of in vitro and vivo studies can be focused on the smaller number of significant studies.

Author Response

Thank you for a favorable evaluation. 

We have added a paragraph with two new references (11-12) on the role of anti-inflammatory drugs in prophylaxis and treatment of cancer. The subject is discussed in lines 76-91. Since we could not find any prospective studies with significant findings, only retrospective studies and meta-analyses, we felt that the subject was not suitable for a table. The referee wrote that the manuscript does a good job with overwhelming the reader with details. A table based on weak studies could compromise that?

Reviewer 3 Report

Title: limit the scope= for example: “Inflammation and digestive cancer”

Lines 36-40: “also mature [36] cells may develop into malignant tumours as seen in the gastric oxyntic mucosa where [37] continuous gastrin hyperstimulation causes the enterochromaffin-like (ECL) cell to proliferate [38] and through stages of ECL cell hyperplasia via neuroendocrine tumour (NET) become [39] come highly malignant carcinomas finally killing the patient”

A schematic picture is welcome to describe gastric interactions between these three cell lines (Cardia? Antrum? Evident visual appearance of these cells and their tissular disposition?) ! It is the central point of your manuscript, and it is far from “reader digest”, even for gastro-enterologist with pathological expertise. So, a picture is welcome. A clear picture will add to the convincing force of this paper.

Line 148 : « Lauren intestinal type of cancers [26]” (replace Laurens by Lauren)

Useful to add this “historical” paper as reference 26, which replaces your reference 107:

Lauren P (1965)
The two histological main types of gastric carcinoma: diffuse and so-called intestinal-type carcinoma. an attempt at a histological classification.
Acta Pathol Microbiol Scand 64:31–49

Line 327: Reference included in the legend of Figure 1: why don’t you refer to if by simple numericalcitation?

               Waldum, HL; Hauso, O.; Sørdal, OF.; Fossmark, R. Gastrin may mediate the carcinogenic effect of Helicobacter pylori infection of the stomach.
               Dig Dis Sci 2015, 60, 1522-1527

Line 331  Table 1: why don’t you include Crohn disease in this table? It would be more in conformity  with your  manuscript. Even if it is a “counter-example” as there are less colic cancer risk whith Crohn disease than with Ulcerative colitis.

General problem:

Obviously, you have already published on the same thema. I agree that you have added many recent publications, so it is an “actualization” of a concept already submitted by your team since at least 13 years. To discard a criticism of plagiarism, it would be useful in the “introduction” to specify that the present manuscript is an “actualization” of a concept developed by your team since YYYY by introducing more recent published material supporting this view.

Author Response

Thank you for a favorable evaluation.

We have done the following changes according to the suggestions: 1. We have changed the title. :2. We have added a new figure 1 and made additions to the former figure 1, now figure 2. Hopefully, this will make the subject easier to grasp.: 3. We have made changes  about the reference to Lauren. :4. We have followed the suggestions and only referred to the numbers in the reference list. : 5. Thank you, we have included Crohn´s disease 6. We have added a phrase in the introduction about our previous publications, lines 59-61.    

Round 2

Reviewer 1 Report

No other Q